# Concerns Related to the Consequences of Pediatric Cannabis Use: A 360-Degree View

**DOI:** 10.3390/children10111721

**Published:** 2023-10-24

**Authors:** Flavia Padoan, Chiara Colombrino, Francesca Sciorio, Giorgio Piacentini, Rossella Gaudino, Angelo Pietrobelli, Luca Pecoraro

**Affiliations:** Pediatric Unit, Department of Surgical Sciences, Dentistry, Gynecology and Pediatrics, University of Verona, 37126 Verona, Italy

**Keywords:** cannabis, adolescent cannabis use, medical uses of cannabis, cannabis short-term and long-term effects, public health initiatives

## Abstract

Cannabis, a plant known for its recreational use, has gained global attention due to its widespread use and addiction potential. Derived from the *Cannabis sativa* plant, it contains a rich array of phytochemicals concentrated in resin-rich trichomes. The main cannabinoids, delta-9-tetrahydrocannabinol (THC) and cannabidiol (CBD), interact with CB1 and CB2 receptors, influencing various physiological processes. Particularly concerning is its prevalence among adolescents, often driven by the need for social connection and anxiety alleviation. This paper provides a comprehensive overview of cannabis use, its effects, and potential health risks, especially in adolescent consumption. It covers short-term and long-term effects on different body systems and mental health and highlights the need for informed decision making and public health initiatives, particularly regarding adolescent cannabis use.

## 1. Introduction

Cannabis is an unconventional plant commonly known for its intoxicating effects. Alongside alcohol and tobacco, it represents one of the most widespread addictions worldwide [1]. This substance is derived from the *Cannabis sativa* plant, belonging to the Cannabaceae family, native to South Asia and Central Asia [2] and, nowadays, extensively cultivated in Africa, Canada, Europe, and the United States [3]. The cannabis plant is rich in phytochemicals, primarily present in resin, within small crystals known as trichomes, located on the surface of the blossoms of mature unfertilized female specimens [4]. Marijuana (MJ) is obtained by drying the leaves and blossoms of the plant, while hashish is produced by drying the resin that accumulates on the leaves. These products are typically smoked, vaporized, or chewed when used as illicit substances; however, they can also be incorporated into pharmaceutical formulations or controlled-dose foods and beverages for therapeutic purposes [5]. The primary compounds found in cannabis are delta-9-tetrahydrocannabinol (THC) and cannabidiol (CBD). The biological effects of phytocannabinoids are facilitated through the G-protein-coupled cannabinoid receptors, CB1 and CB2 [6]. CB1 receptors are predominantly located in the central and peripheral nervous systems and the liver and pancreatic islets [7], while CB2 receptors are more prominently expressed in immune cells [8]. These receptors bind endocannabinoids like anandamide (AEA) and 2-arachidonoylglycerol (2-AG), which play essential roles in regulating various physiological processes, including appetite, pain perception, mood, memory, and inflammation [9]. Regarding exogenous THC, it acts as a partial agonist for CB1 and CB2 receptors, with a higher affinity for the former, which appears responsible for its psychotropic effects [10]. Conversely, CBD exhibits a low affinity for both CB1 and CB2 receptors [11]. CBD is an encouraging cannabinoid, as it has shown promise as a therapeutic agent in preclinical models of central nervous system disorders, including epilepsy, neurodegenerative conditions, schizophrenia, multiple sclerosis, mood disorders, and the central regulation of feeding behavior [12]. Cannabis is the primary illicit substance associated with abuse and dependence development between 12 and 17 years [13,14]. The most common motivations leading adolescents to use the so-called “joint” are the need to connect with peers, self-improvement, and a desire to reduce anxiety related to social contexts [15]. European School Survey Project on Alcohol and Other Drugs (ESPAD) is a repeated cross-sectional multinational survey conducted every four years since 1995, designed to provide nationally representative and comparable data on substance use and other risk behaviors among 16-year-old students in Europe through an anonymous self-administered questionnaire. According to the 2019 ESPAD report [16], the mean lifetime prevalence of cannabis utilization among adolescents in participating nations stands at 16%, demonstrating significant cross-country variation. The nations with the highest cannabis usage rates were Czechia (28%), Italy (27%), and Latvia (26%). Conversely, Nordic countries (the Faroes, Iceland, Norway, and Sweden), Balkan states (Kosovo, North Macedonia, Serbia, Montenegro, and Romania), Cyprus, and Greece all reported rates below 10%. On average, 2.4% of ESPAD students reported initiating cannabis use at age 13 or earlier. Regarding high-risk cannabis use, as indicated by the Cannabis Abuse Screening Test (CAST) results, 4.0% of students across the ESPAD group are susceptible to developing cannabis-related disorders. Notably, the prevalence of cannabis use in the past 12 months was remarkably low in Kosovo, Cyprus, Montenegro, Serbia, and Sweden, despite a higher proportion of users at a high risk of developing cannabis-related issues. Conversely, several countries with the highest rates of past-year use (The Netherlands, Latvia, and Czechia) reported some of the lowest proportions of high-risk users. This implies no direct correlation between cannabis consumption and risky usage. Additional factors, such as the quantities consumed, genetic predispositions, and broader social and cultural determinants, could play a role. These determinants may encompass societal attitudes towards cannabis use, affecting self-assessment of excessive consumption, willingness to discontinue use, recommendations from external sources (e.g., parents or teachers) to cease use, and the emergence of conflicts related to consumption [17,18]. In 2017, the European Monitoring Centre for Drugs and Drug Addiction (EMCDDA) assessed cannabis regulations in Europe, with a particular focus on “recreational” use [19]. Within the European Union (EU), there is no standardized legal framework governing cannabis consumption. While the EU has established some regulations regarding cannabis trafficking offences, individual member states continue to hold primary responsibility for legislating against unauthorized cannabis use and minor possession [20]. Additionally, the legal treatment of cannabis varies across countries; some nations place cannabis in a legal category similar to other drugs, while, in others, penalties for cannabis-related activities are comparatively lower, often influenced by the perceived level of harm the substance may cause [21]. In Europe, no national government currently supports legalizing cannabis sales for recreational purposes. Each European country maintains legal provisions [22,23]. Over recent years, numerous legislative proposals have been introduced in various national parliaments, seeking to reform extant cannabis policies. Furthermore, specific regions and municipalities have initiated localized endeavors to decriminalize or regulate cannabis. In 2012, 11 states in the USA and the sovereign nations of Canada and Uruguay passed legislation that legalized the growth, processing, and use of cannabis for adults [24,25]. The relaxation of laws regarding recreational cannabis use in the USA has led to various outcomes, including a significant reduction in the retail cost of cannabis [24]. In some EU member states (e.g., Italy, Germany, Poland, UK, Finland, Norway, Sweden, etc.), allowances are made for medical use of cannabis. Cannabis can be prescribed as a support to conventional therapies when the latter have failed to achieve the intended outcomes, have resulted in intolerable adverse effects, or necessitate dose escalations that may result in side effects. The medical uses of cannabis and specific medications used in clinical practice are outlined in Table 1, although, in most cases, they refer to studies conducted on adults [19]. Delta-9-tetrahydrocannabinol (THC or Δ9-THC), the primary psychotropic compound in cannabis, acts on numerous neurotransmitter pathways, quickly leading to well-being, euphoria, relaxation, sedation, and motor relaxation [26]. However, chronic use is associated with significant short- and long-term side effects, affecting various biological systems and multiple brain areas [27].

## 2. Methods and Materials

We conducted a nonsystematic review, encompassing the most pertinent research from databases such as PubMed and the Cochrane Library, from January 1932 to September 2023. We focus exclusively on the potential negative effects of acute and chronic cannabis use in adolescence and the short- and long-term consequences on health. We selected manuscripts from various sources, including randomized controlled trials, case reports, reviews, systematic reviews, cohort and case–control studies, and observational studies. The search terms employed included “Cannabis” [all fields], “Adolescent cannabis use”, “cannabis short-term effects” [all fields], “Cannabis long-term effects” [all fields], and “Cannabis and public health initiatives” [all fields].

## 3. Short-Term Effects

Short-term effects include altered perception of time, mood changes, difficulty with thinking and problem-solving, altered senses, impaired movement, short-term memory alterations, and, when taken in high doses, hallucinations, illusions, and psychosis.

### 3.1. Acute Intoxication

One of the primary short-term effects of cannabis consumption is acute intoxication. Per the DSM-5 diagnostic criteria, symptoms should manifest within two hours of cannabis consumption, with no other medical conditions or substance intoxication as the underlying cause. These symptoms include clinically significant problematic behaviors and psychological alterations, such as impaired motor co-ordination, feelings of euphoria, anxiety, a distorted sense of time, compromised critical thinking, and social isolation. These effects tend to arise during or shortly after cannabis use. Additionally, the diagnostic criteria require two or more of the following signs or symptoms within two hours of cannabis use: conjunctival injection, increased appetite, dry mouth (xerostomia), and tachycardia. Notably, a percentage ranging from 2% to 16% of the general population reports experiencing lifetime psychosis-like intoxication effects, such as hallucinations, delusions, or depersonalization. Typically, these effects are of a transient or attenuated nature [37]. Furthermore, several studies have established that simultaneous use of alcohol and marijuana (co-use), prevalent among young adult drinkers, leads to a phenomenon known as “Cross-Fading”, resulting from the combined impact of the two substances on absorption and blood concentration [38]. This cross-fading effect differs from using either substance alone and enhances the sensation of intoxication.

### 3.2. Anxiety Disorder

According to the DSM-5, marijuana use is associated with a significant impact on mental disorders, which can be categorized as “Cannabis Use Disorders”, such as abuse and dependence, and “Cannabis-Induced Disorders”, such as acute intoxication, delirium, psychotic disorder with hallucinations or illusions, anxiety disorder, and other unspecified cannabis-related disorders [39]. The most commonly acute symptoms linked to the use of cannabis are anxiety responses, panic attacks, and agoraphobia [40]. It is important to distinguish these from organically rooted psychiatric disorders such as major depression, bipolar disorder, or paranoid schizophrenia. In particular, cannabis use during adolescence has been identified as one of the main modifiable risk factors for anxiety disorders in adulthood [41]. The likelihood of diagnosing anxiety or mood disorders, whether within the last month or over a person’s lifetime, is roughly twofold higher in individuals with cannabis dependency. Thus, subjects with cannabis dependence have a higher comorbidity with anxiety disorders, between 6.9 and 29%, depending on the disorder [40]. While a direct causal connection between cannabis use and the emergence of anxiety remains uncertain, regular cannabis users tend to display a higher occurrence of anxiety disorders. Additionally, individuals with anxiety disorders demonstrate relatively higher levels of cannabis consumption. “Cannabis-induced delusional disorder” is a syndrome that manifests shortly after cannabis use and is sometimes misdiagnosed as schizophrenia. Typical manifestations include persecutory hallucinations, which can be accompanied by depersonalization, emotional lability with panic attacks, and occasional subsequent amnesia [39].

### 3.3. Suicidal Tendencies

An important aspect of marijuana use on public health is its potential impact on the human mind, particularly in triggering suicidal thoughts. Licanin et al. (2003) [42] discovered that suicidal thoughts were more prevalent among individuals who abused cannabis (50.0%) and those who abused alcohol (36.6%), irrespective of gender or socio-economic status, in comparison to non-abusers. Another study conducted in Bosnia and Herzegovina confirmed this correlation, particularly among those reporting suicidal ideation, where 36.2% were using cannabis [43].

### 3.4. Allergic Reactions

Allergic reactions have been described following exposure to cannabis pollen, marijuana smoke, ingestion of hemp seeds, and direct contact with cannabis plants. Sensitization and allergy to cannabis can result from exposure to allergens specific to *Cannabis sativa* and cross-reactivity with structurally similar foods. Only a limited number of documented IgE-dependent allergic reactions have been reported among illicit drug abusers, making it challenging to establish the actual frequency of this allergy [44,45]. Since cannabis is a plant, individuals sensitized to specific cannabis proteins can experience adverse reactions after consuming various fruits and vegetables due to cross-reactivity. This phenomenon is known as the “cannabis-fruit-vegetable syndrome” [46]. Potential clinical manifestations associated with cannabis pollen exposure include rhinitis, conjunctivitis, contact urticaria, angioedema, and exacerbation of asthma [46,47,48]. Ingestion of hemp seeds and extracts has been associated with gastrointestinal symptoms, such as abdominal cramps, and sporadic cases of anaphylactic shock have been reported [49]. Diagnosis of *Cannabis sativa* allergy relies on patient history and allergen testing. Gathering a history of cannabis consumption can be challenging due to the illegal nature of its use. A recent cross-sectional questionnaire-based study revealed that most asthma and allergic disease patients prefer not to discuss their cannabis use with their doctors and, simultaneously, doctors often fail to inquire about marijuana consumption [50]. Currently, no standardized allergen tests are commercially available. Treatment does not differ from that of common respiratory or food allergies. Prevention through abstinence from consumption represents the optimal management approach. If symptoms occur, they should be treated based on their expression phenotype with second-generation antihistamines, inhaled or oral corticosteroids, antihistamine eye drops, or mast cell stabilizers [51]. A case of successful anaphylaxis treatment using subcutaneous omalizumab administration has been reported in the literature [52]. Several studies investigate allergen-specific immunotherapy with *Cannabis sativa* via intramuscular or subcutaneous routes.

## 4. Long-Term Effects

### 4.1. Neurocognitive Decline in Adulthood

Chronic cannabis use during the vulnerable adolescent period seems to be associated with neurocognitive development deficits in adulthood, leading to a decline in various functional domains [53]. Numerous studies have demonstrated that exposure to marijuana during the crucial period of neurological development in youth has a markedly negative impact on processes such as neuronal maturation, myelination, synaptic pruning, neuronal dendritic plasticity, as well as neurotransmitter system development, exerting a neurotoxic effect on the latter and resulting in long-term irreversible effects [54,55]. Among adolescent cannabis users, lower IQ scores, deficits in verbal learning, mnemonic deficits, attention deficits, and executive function deficits in adulthood have been observed [56,57,58]. These results imply potential anomalies in the hippocampal, subcortical, and prefrontal cortex [54]. Evidence suggests that individuals who begin using cannabis early may be more vulnerable to lasting neuropsychological deficits than those who start later. A study conducted at Duke University in New Zealand illustrated that adolescents who started smoking marijuana during adolescence lost 8 IQ points between the ages of 13 and 38 [53]. The degree and duration of impairment could be influenced by variables like the amount, regularity, duration, and the age at which cannabis use commences, as individuals with more frequent and prolonged use appear to experience more severe and lasting impairment. Furthermore, it has been demonstrated that discontinuing cannabis use does not restore neuropsychological functioning, confirming the neurotoxic effect of this substance [53].

### 4.2. Onset of Psychosis, Anxiety, and Depression

Numerous studies have revealed a connection between persistent cannabis use during adolescence and an elevated risk of developing psychiatric conditions in adulthood. Specifically, exposure to cannabis during youth moderately heightens the likelihood of experiencing psychotic symptoms [1,59,60]. According to Scholer et al. [61], the lifetime prevalence of cannabis-related psychotic symptoms stands at 0.25% for paranoia, 0.15% for the combination of paranoia and hallucinations, and 0.07% for the presence of hallucinations alone. The association between cannabis use and psychosis is underpinned by two hypotheses: the first posits a “cannabis psychosis” or “toxic psychosis”, suggesting that the consumption of high doses of marijuana can lead to the development of a psychotic disorder, which may not manifest without substance use. The second hypothesis links cannabis use in individuals predisposed to the disease to a four times higher risk of symptom onset and exacerbation than non-users [39]. While cannabis use moderately raises the risk of psychotic symptoms in young individuals, its impact is considerably more pronounced in those with a predisposition for psychosis than in those without [62]. Furthermore, cannabis use acts as a predisposing factor for the emergence of symptoms akin to the primary features of dysthymic disorder [39].

### 4.3. Cannabinoid Hyperemesis Syndrome

Several studies have found that individuals with chronic exposure to marijuana frequently seek emergency care for nausea and vomiting [63]. The so-called “Cannabinoid Hyperemesis Syndrome (CHS)” was first described in 2004. It is characterized by cyclic episodes of nausea and vomiting that are relieved by hot baths [64]. Patients often start vomiting profusely, often without warning, and may also experience nausea, sweating, abdominal cramps, polydipsia, and a compulsion to take extremely frequent hot showers [65]. The Rome IV criteria for diagnosing CHS in adults describe cyclic vomiting episodes after prolonged marijuana use over 3 months, with symptom onset at least 6 months prior [66,67]. CHS is typical in males with a history of prolonged marijuana exposure (typically 3–5 times per day) in the preceding two years [67]. According to the literature, 16–30% of cannabis abusers experience CHS, sometimes requiring medical treatment [68,69]. The physiopathological mechanism of cyclic vomiting in chronic MJ smokers is not fully understood. Furthermore, this effect is paradoxical, given the well-established antiemetic properties of cannabis and its derivatives [70]. Cannabis has been shown to slow gastric emptying, potentially promoting nausea and vomiting [71,72]. The impact of prolonged cannabis use on the hypothalamic–pituitary–adrenal axis is a potential contributor to this syndrome [73]. The CB1 and CB2 receptors, widely distributed in the central nervous system, dorsal ganglia, hypothalamus, hippocampus, and cerebellum, as well as in the enteric nervous system and presynaptic ganglia of the parasympathetic nervous system, are responsible for mediating the effects of cannabinoids. They do so by reducing the release of anterior pituitary hormones and increasing corticotropin secretions [74,75,76]. The disruption of normal thermoregulation due to cannabis use, facilitated by CB1 receptors located in the preoptic area and their involvement in the cooling effects of cannabinoids, might provide an explanation for the relief of symptoms experienced by many CHS patients who find relief through compulsive hot baths.

### 4.4. Respiratory System Effects

Chronic marijuana use leads to lung function alterations, lung parenchyma destruction, and, when used in conjunction with tobacco, an increased risk of carcinoma [77,78]. The combustion of cannabis generates a plethora of similar compounds and demonstrates analogous attributes to cigarette smoke. Comparable to the latter, it has been correlated with elevated respiratory symptoms, encompassing persistent cough, sputum production, dyspnea, hoarseness, chest constriction, respiratory distress, and wheezing [78,79]. Although acute marijuana exposure is linked to bronchodilation [79], the co-use of marijuana and tobacco amplifies the susceptibility to respiratory symptoms and chronic obstructive pulmonary disease (COPD) [80]. It also reduces lung parenchymal density on HRCT scans and promotes the development of oropharyngeal and laryngeal carcinoma in individuals under 40 years of age [81]. Marijuana smokers take deeper breaths than tobacco smokers, increasing inspiratory pressure during Valsalva maneuvers and causing barotrauma. This effect, combined with the primary toxic effect of the substance, leads to lower airway emphysematous bullae and secondary spontaneous pneumothorax [82,83]. Moreover, persistent cannabis smoking among young adults induces histopathological modifications in bronchial epithelial cells, encompassing goblet cell hyperplasia, ciliated epithelium loss, and squamous metaplasia [84,85]. Additionally, there is substantiated evidence of cumulative bronchial epithelial harm in individuals who concomitantly consume cannabis and tobacco, as marijuana cigarettes harbor elevated quantities of ammonia, hydrogen cyanide, nitric oxide (NO), and specific aromatic amines, with concentrations surpassing those in tobacco smoke by a factor of 3 to 5 [86]. Additionally, it has been demonstrated that exposure to cannabis products before anesthesia can lead to physiological cardiorespiratory alterations, posing a greater risk of complications [87].

### 4.5. Cardiovascular System Effects

Observational studies have proffered a plausible linkage between enduring marijuana utilization and heightened cardiovascular jeopardy [88]. Consumers of marijuana cigarettes containing tobacco have been shown to have an increased risk of subclinical atherosclerosis in middle age [89]. Furthermore, through sympathetic nervous system activation, marijuana smoking leads to increased heart rate, blood pressure (both systolic and diastolic in the supine position), and forearm blood flow, posing a risk of latent exertional angina in patients with a history of stable angina [90]. Young daily male cannabis users have been found to have an increased risk of myocardial infarction onset, ranging from 1.5% to 3%. This effect is likely due to coronary arterial vasospasm. On the other hand, there have been studies on the beneficial effects of cannabidiol (CBD) [91], a nonintoxicating component of cannabis that is generally well tolerated and has vasodilatory and antioxidant properties. It could potentially be used to treat cardiovascular disorders such as hypertension. Other positive effects of CBD include anti-inflammatory and antiapoptotic effects, which may be beneficial at the cardiac level, protecting against myocardial ischemia-reperfusion injury [92] and diabetes-associated heart disease. Nevertheless, in 2020, the American Heart Association released a scientific statement discussing the increasing concern related to the expanding legalization of medical and recreational cannabis use among adolescents, emphasizing the detrimental impact of cannabis on cardiovascular pathophysiology.

### 4.6. Immunotoxic Effects

There is literature evidence indicating a potential immunotoxic effect of cannabis [93], mediated by the existence of CB1 and CB2 receptors on immune system cells. Endocannabinoids naturally activate these receptors, which are produced and released on demand to modulate immune system activity [94]. By activating the same receptors, exogenous cannabinoids (particularly CBD) act on B lymphocytes, T lymphocytes, monocytes, and microglia. This mechanism results in a reduction in proinflammatory cytokines and an increase in anti-inflammatory ones, leading to immunosuppression through the activation of immune regulatory cells and apoptosis induction [95]. On one hand, the potential therapeutic effects of these substances can be hypothesized, particularly in treating inflammatory diseases such as rheumatoid arthritis [96], IBD, and systemic sclerosis. On the other hand, immune system dysregulation leads to decreased efficacy, increasing susceptibility to infections, particularly respiratory ones. Preclinical studies examining the effect of THC on the immune system of adolescent rats [97,98] have shown that prolonged THC exposure results in a short-term anti-inflammatory effect, which, after an extended period of abstinence, evolves into a paradoxical effect, with increased inflammatory status in both the brain and periphery. These results raise the possibility that exposure to cannabis in human adolescents may lead to increased vulnerability to immune and neuroinflammatory diseases in adulthood [99].

### 4.7. Reproductive System

Chronic cannabis consumption in adolescence has been associated with negative effects on male and female fertility [100]. Cannabinoid receptors have been isolated in the hypothalamus, pituitary gland, ovary, endometrium, testes, and sperm. Marijuana use appears to harm central reproductive processes, including gonadotropin-releasing hormone (GnRH) secretion, follicle-stimulating hormone (FSH) and luteinizing hormone (LH) secretion, ovulation, fertilization, and placentation [101]. Clinically, this translates to an increased risk of delayed puberty or, in documented cases, pubertal arrest [71]. In male cannabis consumers, reduced testicular mass [102] and altered spermatogenesis [103] have been observed. Female consumers have exhibited lengthened menstrual cycles and reduced fertility [104]. Preclinical studies suggest that these toxic effects are primarily attributable to THC and are dose-dependent [105,106].

### 4.8. Type 1 Diabetes Mellitus

Recreational cannabis use may negatively impact metabolic control and self-management in patients with type 1 diabetes. Currently, there is no direct negative effect of cannabis on glucose metabolism [107], but evidence suggests that cannabinoids could induce the death of pancreatic beta cells by directly inhibiting insulin-activated receptors [108]. Additionally, the appetite-stimulating effect of this substance might lead to increased calorie intake and poor disease control [109]. In this context, the psychological aspect of the patient should not be overlooked, especially considering prolonged marijuana use is associated with apathy and decreased motivation. Literature studies show that adolescents with type 1 diabetes report less extensive use of soft drugs than their healthy peers [110]. However, patients with this condition who use marijuana tend to have poorer glycemic control compared to non-users [111], resulting in an increased risk of diabetic ketoacidosis [112]. Consequently, preventing the illegal use of soft drugs should be integral to managing adolescent patients with type 1 diabetes.

## 5. Future Directions

Cannabis use among adolescents is a topic of concern due to its potential short-term and long-term health effects. While cannabis is often perceived as harmless, the evidence presented in this discussion underscores the importance of raising awareness about its risks and implementing effective preventive measures to safeguard the well-being of adolescents. In cannabis research, future directions should prioritize a deeper understanding of the prevalence and incidence of often-overlooked side effects. For instance, there is a need for comprehensive investigations into the long-term impact of chronic cannabis use on the endocrine system. While the immediate implications of cannabis consumption have been extensively investigated, the potential negative repercussions on the endocrine system, such as hormonal imbalances and related health implications, remain underrepresented in the literature. Moreover, it is interesting to explore the consequences of chronic cannabis use during and prior to pregnancy on neonatal outcomes. Research has shown associations between maternal cannabis use and adverse effects on newborns, such as premature birth, hospitalization in neonatal intensive care units, low birth weight, and smaller size for gestational age [113]. There is a dearth of data on the long-term prognosis of children born to mothers who consumed cannabis during and before pregnancy. Investigating the enduring impacts of prenatal cannabis exposure is a possible avenue for future research, as it can provide valuable insights into the potential risks and outcomes that may affect these children as they grow. In addition, another intriguing aspect that requires further investigation is cannabis as a potential allergen. Allergic symptoms associated with cannabis can overlap with symptoms directly caused by its consumption. This poses diagnostic challenges and necessitates further research to understand better the manifestation of allergic reactions to cannabis and how to accurately distinguish them from the direct effects resulting from its use. An additional area warranting thorough investigation is the adverse impact of cannabis on the cardiovascular and respiratory systems. A significant challenge in analyzing these effects lies in the fact that a substantial portion of cannabis users consume it in the form of smoking, often in combination with tobacco. This dual usage pattern complicates the discernment of the individual contributions of cannabis and tobacco to the pathogenesis of side effects. As such, future research should aim to disentangle these intertwined factors to establish a more precise understanding of how cannabis independently affects cardiovascular and respiratory health. Such insights are crucial for delineating the risks associated with cannabis use and informing public health interventions that can address these risks effectively.

## 6. Conclusions

The widespread use of cannabis among adolescents for recreational purposes has raised concerns about its temporary and extended health outcomes. Short-term effects include altered perception of time, mood changes, anxiety, social isolation, suicidal tendencies, depersonalization, delirium, and emotional liability with panic attacks. Furthermore, allergic reactions have been described following cannabis exposure. Prolonged cannabis consumption in adolescence can exert enduring influences on neurocognitive development, culminating in cognitive impairments in adulthood or the manifestation of psychiatric conditions such as anxiety, mood disorders, and psychosis in adulthood. This impact is more pronounced in individuals who start using cannabis early and with higher frequency than those who start to use cannabis later and sporadically. Cannabis is also linked to respiratory effects like tobacco smoke. Persistent cannabis consumption can detrimentally affect pulmonary function, elevate the risk of chronic obstructive pulmonary disease (COPD), and induce histopathological modifications in the bronchial epithelium. Cardiovascular system impacts, including heightened heart rate, increased blood pressure, and an elevated risk of myocardial infarction, are linked to the use of cannabis, both acutely and chronically. Immunotoxic effects of cannabis are becoming evident, with cannabinoids affecting immune system cells through CB1 and CB2 receptors. While this can have therapeutic implications, it may also lead to immunosuppression and increased inflammatory status in both the brain and periphery. Additionally, cannabis use has implications for the reproductive system, affecting both male and female fertility. Cannabis legislation is changing; more and more countries have legalized its use. There has been a rise in the frequency of acute presentations in emergency departments, affecting adult and pediatric patients, attributable to various health issues linked to cannabis use. These concerns encompass psychological distress, vomiting syndromes, and inadvertent poisonings in pediatric cases. Further studies are needed to assess the public health impact of cannabis legalization. In addition, cannabis is frequently perceived as benign by young individuals, highlighting the imperative for pediatricians and general practitioners to understand the immediate and enduring implications of cannabis use thoroughly. They should also engage in educational efforts to raise awareness among teenagers and their families about the potential risks associated with cannabis consumption.

## Figures and Tables

**Table 1 children-10-01721-t001:** Medical uses of cannabis and specific medications used in clinical practice.

Pain relief in conditions characterized by spasticity-related pain unresponsive to standard treatments, such as multiple sclerosis and spinal cord injuries [28,29,30,31].	Nabiximols (THC and CBD 1:1 ratio) Bedrocan (THC 23% and CBD < 1%)
Pain management in cases of chronic pain, particularly neurogenic pain, when treatment with non-steroidal anti-inflammatory drugs, corticosteroids, or opioids has demonstrated inefficacy [30].	Nabiximols (THC and CBD 1:1 ratio) Nabilone (Synthetic cannabinoid derivate mimicking THC)
The alleviation of nausea and vomiting induced by chemotherapy, radiotherapy, and anti-HIV therapies, which are unattainable through conventional treatments, is attributed to the antiemetic and antikinetic effects [29,30].	Nabilone (Synthetic cannabinoid derivate mimicking THC) Dronabinol (synthetic form of Δ^9^-THC)
Providing the capacity to stimulate appetite in conditions such as cachexia, anorexia, appetite loss in cancer patients or individuals with AIDS, and anorexia nervosa, which remains unattainable through conventional treatments [29].	Dronabinol (synthetic form of Δ^9^-THC)
The hypotensive effect in cases of glaucoma that do not respond to traditional treatments is observed [30].	CBD derivatives
Reduction in involuntary movements in Tourette syndrome, which cannot be achieved with standard treatments [32].	Dronabinol (synthetic form of Δ^9^-THC)
Reduction in seizure frequency associated with Dravet syndrome, Lennox-Gastaut syndrome, and tuberous sclerosis complex [33,34].	Highly purified CBD(>98% *w*/*w*)
Alleviation of Parkinson-Disease-related tremor, anxiety, pain, and improvement of sleep quality and quality of life [35,36].	CBD derivatives

## Data Availability

Not applicable.

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
