# Peer review of "Concerns Related to the Consequences of Pediatric Cannabis Use: A 360-Degree View"

_children, 2023, doi:10.3390/children10111721_

Round 1

Reviewer 1 Report

Page 2. Lines 87-89.

Delta-9-tetrahydro- 87 cannabinol (THC or Δ9-THC), the primary psychoactive compound in Cannabis, acts on 88 numerous neurotransmitter pathways, quickly leading to well-being, euphoria, relaxa- 89 tion, sedation, and motor relaxation [22].

Please note that THD is psychotropic not psychoactive.

Page 2. Table 1.

Check please the title of table. Also, medical uses must be referenced in the table or, if is the case, include the annotation of the reference of which information was extracted. Authors might include a column with the specific mediation for each disease (THC, CBD, THC/CBD medications).

In all text, Cannabis sativa in italic.

Page 7. Future directions.

The authors should be more specific about the areas of research that need to be addressed or those that have not been studied to help understand the impact of cannabis use on children and adolescents.

Page 8. Conclusion

Conclusions should not be supported by references, on the other hand, the conclusion is too general, authors might be most precise.

Author Response

(see attached)

Reviewer 2 Report

Thank you very much for giving me the opportunity to review this manuscript. In this paper, the authors take a comprehensive view of cannabis use in young people. And indeed, they have made a very comprehensive description, with a very correct and detailed explanation of the effects of cannabis on the body, at the physiological level, as well as its impact on mental health and at the social level in young people. I really enjoyed reading your manuscript.

Perhaps the biggest weakness I detected in your work is that I don't know how you arrived at these data. That is, the methodology is missing. And as a minor comment, I found errors in the bibliography. 

Author Response

(see attached)

Reviewer 3 Report

Manuscript IDchildren-2623064

Title: Cannabis use among young individuals: a 360-degree view

Children

The manuscript discusses the use of cannabis in young individuals and its short- and long-term effects on the periphery and central nervous system. Specifically, it provides an overview on the health risks associated with it. This well-written review has a defined outline. The overall structure is logical. However, a main caveat of this review is that it does not clearly distinguish between adolescence use and adult use. This is critical, as the review seems to provide one prominent perspective (negative effects of cannabis) without a critical analysis of the literature, which is specifically important due to the recent policy changes in Europe as well as in the world, e.g. the US. Overall, this article provides a review of an important and timely topic but a more critical analysis should be provided with additional discussion points, as well as a complete focus on cannabis use among young adolescence. Specific suggestions are as follows:

Major Comments:

-       the review lacks a clear distinction between adolescence use and adult use; no studies should be cited in his review that are based on adult use (meaning when cannabis use started in adulthood), except for an introductory paragraph that may briefly characterize cannabis use in adults; this includes Table 1 unless it is solely based on medical use of cannabis in adolescence, same for allergic reactions for short-term effects, the citations provided do not focus on adolescence but just general allergies that can develop 

-       it seems the that the review wants to focus on the negative effects of cannabis exclusively, if that is the case the title of the manuscript should change

-       when describing the different effects (short-term and long-term) it would be important to know if all adolescence users will experience it or how prevalent it is, e.g. short-term effect allergic reactions, suicidal tendencies ïƒ  the main problem is the lack of providing a critical analysis, further how is it different from adult use, do these events happen more in young adolescence or same in all ages? All of the sections are lacking this type of information. 

-       should describe the policy changes that have happened over the last couple of years, hardly any recent studies are mentioned, reference list does not include a single paper from 2023, 1 paper from 2022, 7 papers from 2021

-       line 43: information on CBD being able to reduce side effects of THC ïƒ  there is conflicting literature out there, can’t always reverse it especially if the ratio of CBD:THC is low

Minor Comments:

-       line 69,70 “That suggests no simple and direct relation exists between cannabis use and risky use, with potential influences of other factors, such as quantities used and broader social and cultural factors.” ïƒ genetic factors should be mentioned here as well

-       Table 1, other diseases are missing, eEpilepsy, Parkinson’s etc.

-       Line 105, “… more of the following signs or symptoms that develop …” ïƒ  should read: “… more of the following signs are symptoms that develop …”

-       Add a reference to this statement, lines 166-170: “Numerous studies have demonstrated that exposure to marijuana during the crucial period of neurological development in youth has a markedly negative impact on processes such as neuronal maturation, myelination, synaptic pruning, neuronal dendritic plasticity, as well as neurotransmitter system development, exerting a neurotoxic effect on the latter and resulting in long-term irreversible effects.”

-       Add a reference to this statement, lines 170-172: “Among adolescent cannabis users, lower IQ scores, deficits in verbal learning, mnemonic deficits, attention deficits, and executive function deficits in adulthood have been observed.”

Author Response

(see attached)

Round 2

Reviewer 2 Report

The authors have addressed the issues raised in the review. have improved the final quality of the manuscript. I congratulate them on their work.

Reviewer 3 Report

The authors addressed all my concerns